# Multi-Objective Optimization of the Process Parameters in Electric Discharge Machining of 316L Porous Stainless Steel Using Metaheuristic Techniques

**DOI:** 10.3390/ma15196571

**Published:** 2022-09-22

**Authors:** Himanshu Singh, Praful Patrange, Prateek Saxena, Yogesh M. Puri

**Affiliations:** 1Department of Mechanical Engineering, Visvesvaraya National Institute of Technology, Nagpur 440010, Maharashtra, India; 2School of Mechanical and Materials Engineering, Indian Institute of Technology, Mandi 175005, Himachal Pradesh, India

**Keywords:** porous SS316L, sintering, electric discharge machining, optimization

## Abstract

Electric discharge machining is an essential modern manufacturing process employed to machine porous sintered metals. The sintered 316L porous stainless steel (PSS) components are widely used in diverse engineering domains, as interconnected pores are present. The PSS material has excellent lightweight and damping properties and superior mechanical and metallurgical properties. However, conventional machining techniques are not suitable for porous metals machining. Such techniques tend to block the micro-pores, resulting in a decrease in porous materials’ breathability. Thus, the EDM process is an effective technique for porous metal machining. The input process parameters selected in this study are peak current $\left(I_{p}\right)$, pulse on time $\left(T_{on}\right)$, voltage (V), flushing pressure $\left(f_{p}\right)$, and porosity. The response parameters selected are material removal rate (MRR) and tool wear rate (TWR). The present work aims to obtain optimum machining process parameters in the EDM of porous sintered SS316L using two meta-heuristic optimization techniques, i.e., Teaching Learning-Based Optimization (TLBO) and Particle Swarm Optimization (PSO) algorithms, to maximize the MRR and minimize the TWR values. In the case of PSS having a 12.60% porosity value, PSO and TLBO algorithms give same optimum machining parameters. However, for PSS having an 18.85% porosity value, the PSO algorithm improves by about 5.25% in MRR and by 5.63% in TWR over the TLBO. In the case of PSS having a 31.11% porosity value, the PSO algorithm improves about 3.73% in MRR and 6.46% in TWR over the TLBO. The PSO algorithm is found to be consistent and to converge more quickly, taking minimal computational time and effort compared to the TLBO algorithm. The present study’s findings contribute valuable information in regulating the EDM performance in machining porous SS316L.

## 1. Introduction

The thermal sintering technique is employed to manufacture 316L porous stainless steel (PSS). Sintering of the metal powders is performed inside a controlled environment, at a temperature below the melting point of the base metal [1]. PSS has low cost, easy availability, good workability, high fatigue life, fracture toughness, low density, large specific surface area, excellent energy absorbtion properties, electrical conductivity, weldability, and ductility of metallic materials [2]. Porous metals such as titanium and titanium alloys, cobalt chrome, nitinol shape memory alloys, and stainless steel 316L are generally employed for biomedical and membrane filtration applications [3]. Among these materials, SS316L porous metals have become a suitable candidate for bio-materials, which can increase bone fixation and are extensively used in hip and knee replacement surgery [4]. The PSS has excellent lightweight and damping properties and superior mechanical and metallurgical properties. Porous SS316L metal membranes are employed to filtrate a gas mixture of CO and $C_{4}{\mathrm{NiO}}_{4}$ at temperatures up to 250 °C and 8 MPa for 4 months with excellent filtration characteristics [5]. The materials selection and different manufacturing techniques for various industrial applications of porous metal membranes, such as membrane contractor, membrane bioreactor, catalytic metal membranes, etc., are critically reviewed by Singh et al. [6].

The presence of interconnected pores leads to ventilation and breathing characteristics in porous SS316L. The presence of such excellent characteristics of PSS are highly suitable for metal membrane and die casting [7]. Furthermore, due to the presence of interconnected pores, the evacuation of air is easily possible. However, conventional machining process such as turning, milling, and grinding deteriorate the pores’ interconnectivity properties. Hence, the breathing capacity of the PSS diminishes. This phenomenon decreases the mechanical and metallurgical characteristics of PSS. Thus, non-conventional machining technology is recommended to preserve the properties of porous SS316L. Furthermore, to reduce the thickness of porous metal membranes and generate complex three-dimensional geometries with higher manufacturing efficiency and good surfaces, EDM machining is essential. Wang et al. [8] have investigated the influence of porosity and pore size of ASISI 304 PSS on micro-EDM machining characteristics. The porous substrates are employed to enhance biological fixation on orthopedic implants. In the case of fixation of sensors and other devices, the drilling of PSS is essential. Hence, a high surface finish and accuracy of the machined surface are desired. These features can easily be generated by the EDM machining process [9]. Kumar et al. [10] have worked on multivariable optimization in EDM machining AISI 420 stainless steel with a Taguchi-grey technique. Sanjeev et al. [11] have investigated single optimization in EDM machining of stainless steel 316L using Taguchi design. Suresh et al. [12] have employed a response surface methodology for the micro-EDM machining of stainless steel 316L. The genetic algorithm is used for single-objective optimization to achieve higher MRR values.

As there is very little literature available on the EDM of PSS, it is essential to analyze the EDM performance in machining porous SS316L to enhance the MRR and reduce the TWR values. The EDM of porous metal requires optimal process parameters, which significantly benefits manufacturing industries in terms of improved product quality, reduced machining cost, enhanced productivity, etc. Sahoo et al. [13] have experimented on high carbon, high chromium steel, as a workpiece and brass as wire electrode material. They have designed the experiments using Taguchi $L_{9}$ orthogonal array, and machining process parameters are optimized by employing multi-objective optimization by ratio analysis (MOORA) method. Nguyen et al. [14] used the Taguchi Data Envelopment Analysis based Ranking (DEAR)-based multi-criteria decision-making technique to obtain optimal EDM process parameters in machining silicon-based steel with a low-frequency vibration assisted EDM process. Bhiksha et al. [15] have investigated the effect of graphite powder concentration in EDM machining on Ti-6Al-4V alloy. They have used Grey relational analysis for the multi-response optimization of EDM process parameters. Pratap et al. [16] performed the EDM machining of Inconel-X 750. They employed an approach integrating Weightage principal component analysis using Taguchi theory (WPCA-Taguchi).

Nature-inspired heuristic optimization techniques have been shown to be better than deterministic methods and are extensively used. Hence, meta-heuristic optimization techniques are extensively utilized to improve the desired manufacturing process in modern industries [17]. These techniques solve numerous complex, multimodel, and large-dimensional or discontinuous problems and deliver acceptable solutions to complicated problems. The results obtained from such techniques are found to produce solutions that are improved compared to deterministic techniques [18]. These nature-inspired meta-heuristic techniques, such as genetic algorithm (GA), are based on Darwin’s theory of biological evolution, i.e., survival of the strongest. The artificial bee colony (ABC) is inspired by the collective behavior of social insect colonies and other animal societies. Ant colony optimization (ACO) is based on stigmergy and foraging for food sources [19]. In meta-heuristic optimization problem computation, the target is to obtain the global optima. However, this optimum value can only be estimated by forming a fitness function curve using the regression equation of the objective function. In the case of traditional optimization techniques, the formation of the fitness function curve is not attempted. Hence, there is a need for a proper fitness curve so that the local and global optima regions can easily be determined feasibly.

There is limited research published on the application of TLBO and PSO in non-conventional machining processes, especially in the EDM processing of porous metals, and it is yet to be explored. Therefore, the present work aims to obtain the optimum machining process parameters in the EDM of porous sintered SS316L by using the TLBO and PSO algorithms to maximize the MRR and minimize the TWR values. In addition, analysis of variance analysis (ANOVA) is performed to determine the influence of different EDM process parameters and different porosity values of sintered porous SS316L on MRR and TWR values. The final optimized results using two intelligent algorithms, TLBO and PSO, were further analyzed comparatively. The present study’s findings contribute valuable information in regulating the EDM performance in machining porous SS316L.

## 2. Materials and Methodology

### 2.1. Selection of Work Material and Electrode Material

The exploratory investigations in this study are performed on a sinker-type EDM. The model of this sinker-type EDM machine used in this study is ELECTRONICA 500 × 300 ZNC. The workpiece material selected in this study is porous SS316L. Experiments are carried out on this porous substrate using die-sinking EDM. The chemical composition of the SS316L metal powder is shown in Table 1. Copper is used as an electrode material with high electro- and thermo-physical properties, which make copper an ideal material for EDM processes [20]. A 6 mm copper electrode is employed for machining up to a depth of 2 mm. The dielectric used in the experiment is industrial-grade EDM oil (Grade 30).

### 2.2. Sintering of SS316L Metal Powder

Sintering experiments are conducted on a tubular vacuum furnace of MTI Corporation used for experimentation, as shown in Figure 1. Turbo-molecular pump (TMP TV551, Navigator) calibrated at current (I = 1.0 A), power (P = 21 W), and angular velocity (42 RPM) at 29 °C is employed to generate a vacuum pressure of $10^{-5}$ mbar inside the tubular domain. Metal powder is mixed with 1 wt.% of methocyl binder. The binder provides the sticking action between metal particles. The sodium chloride powder is used as a pore is generated. Three different compositions of NaCl are utilized, i.e., 10 wt.%, 20 wt.%, and 30 wt.%. The green sample is compacted at a pressure of 250 MPa. The prepared green samples are further sintered at the optimum sintering parameters, i.e., 1050 °C, 5 °C/min, and 45 min holding time [21].

### 2.3. Input Process Parameters

The input process variables selected in this research are pulse on-time ($T_{on}$), flushing pressure, peak current ($I_{p}$), voltage (*V*), and porosity (ϵ). The other processing variables are kept constant throughout all the experimental runs. The EDM input process parameters are selected in such a manner so that there is no phenomenon of pore blockage on the PSS surface. Three levels are selected for every factor. The levels were selected such that a change in the level of an input parameter causes a significant change in the response parameter. The input variables selected in the experiment are given in Table 2. The constant values of process parameters are given in Table 3.

### 2.4. Experimental Design

In this work, the Taguchi orthogonal array is used to select a combination of various input parameters with different levels. This orthogonal array is one of the types of fractional factorial design [22]. $L_{27}$ orthogonal array is used in this study, as shown in Table 4. The orthogonal arrays are properly balanced, and this method ensures that all levels of all input components are examined equally. This equal attention is owing to the fact that all of the input parameters may be examined individually. As a result, the influence of one input component has no effect on the evaluation of other input factors. To ensure the reliability of the experimental results, each experiment was replicated twice. *MRR* is estimated by the volume of removed material per second. The porosity of the PSS material has been included in calculating *MRR* values. It is the most important machining characteristic and should be as high as possible. This value can be calculated as given by Equation (Equation 1) [8].
(1)$$MRR=\frac{\Pi D^{2}H\left(1-p\right)}{4T}$$
where *D* is the diameter of the hole or copper electrode (6 mm), *H* is the depth of machining (2 mm), *p* is the porosity value (%), and *T* is the machining time (s). The unit of *MRR* obtained is mm${}^{3}$/s. *TWR* is calculated by considering the electrode’s initial and final weight after the machining process. This value should be as low as possible. This value can be calculated as given by Equation (Equation 2).
(2)$$TWR=\frac{\left(w_{1}-w_{2}\right)}{T}$$
where $w_{1}$ is the initial weight of electrode (mg), $w_{2}$ is electrode weight after machining (mg), and *T* is the machining time. The unit of *TWR* obtained is mg/s.

## 3. Optimization Algorithms

In almost all design, analysis, and manufacturing-related problems, the main aim is to determine the upper and lower bounds of the objective functions. Thus, optimization means acquiring the most satisfactory conceivable outcomes under specific events. For example, in most manufacturing processes, the main aim is to improve machining efficiency and reduce machining costs effectively. Hence, meta-heuristic optimization techniques are extensively utilized to improve the desired manufacturing process in modern industries. These techniques solve numerous complex, multimodel, and large-dimensional or discontinuous problems and deliver an acceptable solution to complicated problems to be solved through conventional approaches. TLBO and PSO meta-heuristic optimization techniques are employed in the present work.

### 3.1. Teaching–Learning-Based Optimization (TLBO)

TLBO is a stochastic population-based technique proposed by Rao et al. [18]. The algorithm simulates a teacher’s and students’ capacity to teach and study in a classroom. The algorithm’s two most important elements are the teacher and the learners, who characterize two main modalities of teaching: learning from the instructor (known as the teacher phase) and learning from other learners (known as the learner phase). The TLBO technique considers the outcomes in terms of student performance or scores, which are dependent on the characteristics of the instructor. A high-quality teacher is typically thought of as a well-educated person who instructs students to achieve better scores. Furthermore, students will benefit from their interactions with one another, which helps them improve their performance. The TLBO approach is a population-based technique. In this approach, a population of learners is represented. Various subjects presented to learners are considered diverse design variables, with a learner’s outcome corresponding to the optimization problem’s “fitness” value. The instructor is seen to be the best option among the overall population. The TLBO is separated into the “Teacher Phase” and the “Learner Phase”, as shown in Figure 2. The operation of both phases is described below.

#### 3.1.1. Teacher Phase

The first phase of TLBO is the teacher phase, during which a teacher instructs students. The teacher selects the best student (solution) in the classroom (starting population) with the most information (fitness). As a result of the teacher increasing the class’s average score, the student’s average score also rises. The ideal instructor should be able to upgrade the student’s knowledge up to their level of expertise. However, because of learners’ quality, a teacher can only enhance the mean of a classroom by a particular amount, even at their best. This occurrence is sporadic and is influenced by a variety of circumstances.

#### 3.1.2. Learner Phase

The second part of TLBO is the learner’s phase, in which students obtain new information through effective interaction between individuals, such as group discussions, formal communications, and demonstrations. It aids in the improvement in every student’s grades, consequently enhancing the overall mean of the entire class. If and only if the other student is much more informed will a learner learn from them. The leaner system is achieved by selecting students randomly. Learners are chosen at random in this phase. Applying random selection for learner modification, on the other hand, would result in a significant loss of engagement with learners who have more excellent information. Due to the random nature of selection, learners with higher expertise may not engage in interactions. As a result, in the suggested TLBO, the learner is based on the available interaction and complete engagement, as shown in Figure 2.

### 3.2. Particle Swarm Optimization (PSO)

Particle Swarm Optimization PSO is a meta-heuristic technique. This optimization technique is applied when problems are non-linear and mixed-integer in nature or even when the problem is a black-box optimization problem. PSO has been extensively used in design, manufacturing, computer science, decision sciences, and social sciences. This technique models the social behavior of bird flocking or fish schooling. Each particle/bird has a position and velocity associated with it in this analysis. Particles change their position by adjusting their velocity to seek food, avoid predators, and identify optimized environmental parameters. A significant difference between TLBO and PSO is that each particle memorizes the best location identified by it. It is like keeping a record of one’s own best outcomes and identifying by them. The particle will have its position in memory and keep exploring the search space, but it memorizes its best location. Particles communicate the information regarding the best location explored by them. From this best location, i.e., the best the location of the individual, the particle can be located, which would be the global best. The velocity of the particles is modified by using the flying experience of the particle, as every particle has a velocity associated with it and the flying experience of the group [23]. The flow diagram of the PSO algorithm is given in Figure 3. The following steps are operated for the PSO optimization process.

Parameter limits are selected between the lower and higher values.The particle velocity created is randomly selected between the particle’s higher and lower values.The value of the objective functions is calculated.At the new particle position, the values of the functions are again calculated.The procedure is repeated until the final solution has been achieved.

### 3.3. Multi-Objective Optimization

The TLBO and PSO algorithms perform multi-objective optimization in two ways: a priori and a posteriori. The a priori technique is applied in the present study. The two different response factors are MRR and TWR. To solve a multi-objective problem, the function is normalized. The primary goal is to increase the MRR while minimizing the TWR. The single objective optimized values of both functions were first determined using regression equations, i.e., Equations (Equation 2) and (Equation 3). Then, the single objective optimized value was utilized to produce the regression equation for multi-objective optimization, i.e., maximizing MRR and reducing TWR. Equation (Equation 3) calculates the normalized multi-objective function (Z) using the a priori technique [24].
(3)$$Max.\left(F\right)=w_{1}\frac{MRR}{MRR_{max}}-w_{2}\frac{TWR_{min}}{TWR}$$
where $w_{1}$ and $w_{2}$ are the weights assigned to MRR and TWR, i.e., between 0 to 1. Any values assigned to MRR and TWR imply their relative importance and find a set of decision variable as an optimal solution.

## 4. Results and Discussion

### 4.1. Formulation of Mathematical Model

The EDM response factors have been correlated with the different machining parameters obtained by using the Taguchi design of the experiment to obtain generic mathematical equations. Minitab 19 is employed for developing the regression model for the responses, MRR, and TWR for EDM process parameters $\left(T_{on}\right)$, $\left(I_{p}\right)$, (V), $\left(f_{p}\right)$, and porosity. Thus, the generated mathematical models are described below: The regression equations of MRR values for 12.69%, 18.85%, ad 31.11% porosity are described by Equations (Equation 4), (Equation 6), and (Equation 8), respectively. The regression equations of TWR values for 12.69%, 18.85%, and 31.11% porosity are described by Equations (Equation 5), (Equation 7), and (Equation 9), respectively.
(4)$$MRR=0.0320+0.0057T_{on}+0.0112I+0.0029V-0.0141f_{p}$$
(5)$$TWR=-0.0336+0.0170T_{on}+0.0190I-0.0026V-0.0356f_{p}$$
(6)$$MRR=0.0232+0.0047T_{on}+0.0098I+0.0024V-0.011f_{p}$$
(7)$$TWR=-0.0303+0.0176T_{on}+0.0187I-0.0035V-0.0389f_{p}$$
(8)$$MRR=0.0157+0.0043T_{on}+0.0091I+0.0013V-0.0144f_{p}$$
(9)$$TWR=-0.033+0.0175T_{on}+0.0181I-0.0037V-0.0459f_{p}$$

#### 4.1.1. *R* − *sq* Determination Coefficients

The ANOVA analysis provides one very useful factor, i.e., R−sq [25]. This factor is defined as the ratio of the sum of the square of the calculated answers (corrected average) to the sum of the square of the measured answers (corrected average).

#### 4.1.2. Absolute Average Deviation (*AAD*)

An experimental data set’s average absolute deviation (*AAD*) is the average of the absolute deviations from the central data points. This factor informs about the average manipulation error; the following expressions can define *AAD* [26].
(10)$$AAD=\frac{1}{P}\sum_{i=1}^{p}\frac{Y_{iexp}-Y_{itheo}}{Y_{iexp}}$$
where $Y_{exp}$ is the experimental value and $Y_{theo}$ is the calculated response starting from the model for an experiment *i*; *p* refers to the total number of experiments.

#### 4.1.3. BIAS Factor (BF)


(11)
$$Bf=10^{B}$$


The BIAS *B* is given by the relation:(12)$$B=\frac{1}{n}\sum log\left(\frac{Y_{theo}}{Y_{exp}}\right)$$
where $Y_{exp}$ is the experimental value and $Y_{theo}$ is the calculated response starting from the model for an experiment *i*; *n* refers to the total number of experiments [27].

### 4.2. Parametric Analysis for MRR

The main effect plot for pulse on time $\left(T_{on}\right)$, current $\left(I_{p}\right)$, voltage (V), flushing pressure $\left(f_{p}\right)$, and porosity (ϵ) on MRR are shown in Figure 4. It is noteworthy that data means determine an individual factor’s effect. As mentioned in the same figure, the MRR value is directly proportional to the pulse on time $\left(T_{on}\right)$; i.e., when the pulse on time is increased from 5 μs to 10 μs, the MRR value increases from 0.1486 mg/s to 0.1680 mg/s. Upon further increasing the pulse on time from 10 μs to 15 μs, the MRR values increase from 0.1680 mg/s to 0.1979 mg/s. It can be concluded that MRR values improve upon increasing the pulse on time $\left(T_{on}\right)$. Similarly, when the current values are increased, the MRR value increases accordingly. This phenomenon is because a high current value increases the intensity of the spark striking the PSS workpiece. Therefore, high material erosion occurs. Thus, when the current value increases from 2 A to 6 A, the MRR value increases from 0.1486 mg/s to 0.1680 mg/s. Upon further increasing the current value from 6 A to 10 A, the MRR value increases from 0.1680 mg/s to 0.1979 mg/s. In the EDM process, voltage plays an important role. A high value of the voltage generates a high sparks-intensity value. Therefore, more erosion of the PSS workpiece occurs, resulting in high MRR values. Hence, when the voltage value increases from 15 V to 20 V, the MRR value increases from 0.1574 mg/s to 0.1775 mg/s. There is significantly less increment in the MRR value. Upon increasing the voltage value from 20 V to 25 V, the MRR value increases from 0.1775 mg/s to 0.1797 mg/s. There is significantly less increment in the MRR value. In contrast, flushing pressure is also an important EDM process parameter. High-pressurized fluid removes the debris present between the electrode and the workpiece material. Upon increasing the flushing pressure value from 0 kg/cm${}^{2}$ to 0.5 kg/cm${}^{2}$, the MRR value decreases from 0.1816 mm${}^{3}$/s to 0.1646 mm${}^{3}$/s. Upon further increasing the flushing pressure value from 0.5 kg/cm${}^{2}$ to 1 kg/cm${}^{2}$, the MRR values increases from 0.1646 mm${}^{3}$/s to 0.1683 mm${}^{3}$/s. Such a significant increase in the MRR value may be due to increased fluid turbulence between the workpiece and electrodes, which effectively removes the debris. The present analysis shows the EDM process on three porous SS316L materials with different porosity values. Upon increasing the porosity value from 12.69% to 18.85%, the MRR value decreases from 0.2078 mm${}^{3}$/s to 0.1720 mm${}^{3}$/s. Further, upon increasing the porosity value from 18.85% to 31.11%, the MRR value significantly decreases from 0.1720 mm${}^{3}$/s to 0.1348 mm${}^{3}$/s. This phenomenon is because, in the case of a higher porosity value, the contact between the electrode and the effective volume of PSS is lower, so less material removal occurs, leading to a reduction in MRR value in the case of an increasing porosity value.

To calculate the significant factor and relative importance of each EDM process parameter, ANOVA was conducted. The ANOVA analysis table for MRR values is presented in Table 5. Factors that have *p*-values less than 0.05, indicate that factor is significantly acceptable [28]. The percentage contribution of each process parameter is calculated in the same table. The peak current $\left(I_{p}\right)$ is the most significant factor, and this factor has a 41.7441% contribution to MRR values. The PSS porosity (ϵ) value is in second rank, making a 33.1002% contribution. The pulse on time $\left(T_{on}\right)$ is in third rank, making a 15.5972% contribution. Finally, voltage (V) is in fourth rank, making a 3.1960% contribution. The flushing pressure is a minor significant factor, making a 1.1252% contribution. Thus, peak current $\left(I_{p}\right)$, pulse on time $\left(T_{on}\right)$, porosity (ϵ), and voltage are the most significant parameters for the MRR response. The accuracy and validation of the developed statistical models are further examined by finding the coefficient of determination (R-Sq) for MRR values [29]. The analysis of variance given R-Sq for the MRR response is 94.76%, as shown in Table 6. This numerical value is close to 95%. The BIAS factor is equal to the unit, and the AAD (Absolute Average Deviation) equals zero for MRR response. Therefore, the developed mathematical models are highly accurate and regarded as valid.

### 4.3. Parametric Analysis for TWR

The main effect plot for the pulse on time $\left(T_{on}\right)$, current $\left(I_{p}\right)$, voltage (V), flushing pressure $\left(f_{p}\right)$, and porosity(ϵ) on TWR are shown in Figure 5. Data means define an individual factor’s effect. As mentioned in the same figure, the TWR value is directly proportional to the pulse on time $\left(T_{on}\right)$; i.e., when the pulse on time increases from 5 us to 10 us, the TWR value increases from 0.0796 (mg/s) to 0.1686 mg/s. On further increasing the pulse on time from 10 us to 15 us, the TWR values increase from 0.1680 mg/s to 0.2539 mg/s. It can be concluded that TWR values improve when the pulse on time $\left(T_{on}\right)$ is increased. Similarly, upon increasing the current values, the TWR value consequently increases. This phenomenon is because a high current value increases the intensity of the spark striking the copper electrode material. Therefore, high material erosion occurs. Thus, increasing the current value from 2 A to 6 A, the TWR value increases from 0.08470 mg/s to 0.1837 mg/s. Upon further increasing the current value from 6A to 10 A, the TWR value increases from 0.1837 mg/s to 0.2338 mg/s. In the EDM process, voltage plays an important role. A high value of voltage generates a high sparks intensity value. Therefore, more erosion of the copper electrode occurs, resulting in high TWR values. Hence, when the voltage value increases from 15 V to 20 V, the TWR value decreases from 0.1804 mg/s to 0.1747 mg/s. There is significantly less decrement in the TWR value. Increasing the voltage value from 20 V to 25 V reduces the TWR value from 0.1747 mg/s to 0.1472 mg/s. In contrast, flushing pressure is also an important EDM process parameter. High pressurized fluid removes the debris present between electrode and workpiece material. Upon increasing the flushing pressure value from 0 kg/cm${}^{2}$ to 0.5 kg/cm${}^{2}$, the TWR value decreases from 0.1991 mg/s to 0.1441 mg/s. On further increasing the flushing pressure value from 0.5 kg/cm${}^{2}$ to 1 kg/cm${}^{2}$, the MRR value increases from 0.1441 kg/cm${}^{2}$ to 0.1590 kg/cm${}^{2}$. Such a significant increase in the TWR value may be due to increased fluid turbulence between the electrodes and the workpiece, efficiently removing the debris and carbon particles sticking to the electrode surface. Hence, the TWR value significantly increases. In the present analysis, the EDM process ws performed on three porous SS316L materials with different porosity values. Upon increasing the porosity value from 12.69% to 18.85%, the TWR value decreases from 0.1801 mg/min to 0.1688 mg/min. Further, upon increasing the porosity value from 18.85% to 31.1%, the TWR value significantly decreases from 0.1688 mg/min to 0.1533 mg/min. This phenomenon is because, in the case of a higher porosity value, the contact between the electrode and the effective volume of PSS is less, so less material removal occurs, leading to a reduction in TWR value upon increasing the porosity value.

To calculate the significant factor and relative importance of each EDM process parameter, an analysis of variance (ANOVA) was conducted. The ANOVA analysis table for TWR values is presented in Table 7. Those factors, which have *p*-values less than 0.05, indicate that the factor is significantly acceptable [28]. The percentage contribution of each process parameter is calculated in the same table. The pulse on time $\left(T_{on}\right)$ is the most significant factor, and this factor has a 51.7044% contribution to TWR values. The peak current $\left(I_{p}\right)$ value is in second rank, making a 37.8642% contribution. The flushing pressure is in third rank, making a 2.7405% contribution. Finally, the voltage (V) is in fourth rank, making a 1.8760% contribution. The porosity value (ϵ) is a minor significant factor, making a 1.2218% contribution. Thus, pulse on time $\left(T_{on}\right)$ and peak current $\left(I_{p}\right)$ are the most significant parameters for TWR response. The accuracy and validation of the developed statistical models are further examined by finding the coefficient of determination ( R-Sq) for TWR values [29]. The analysis of variance given R-Sq for MRR response is 95.41%, as shown in Table 8. This numerical value is greater than 95%. The BIAS factor is equal to the unit, and the AAD (absolute average deviation) equals zero for TWR response. Therefore, the developed mathematical models are highly accurate and regarded as valid.

Although graphical judgment is the most intuitive means of effect consideration, the inferences made based on it are not accurate and thus cannot be reliable. However, the plot of factor effects is only comparatively valid. Before any presumptions can be made, normality error, variance consistency, etc., must be checked. Figure 6 and Figure 7 show the normal plot of residuals for MRR and TWR, respectively. A normal probability plot is just a graph of the cumulative distribution of the residuals on a normal probability paper. The Anderson–Darling (AD) statistic is used here to check the normal distribution of the residuals. As shown in Figure 6 and Figure 7, the *p*-value calculated based on AD statistics are 0.820 and 0.061 [30]. These values are higher than the α-level of confidence (0.05), and the error normality is considered to be valid.

### 4.4. Optimization Using TLBO and PSO Algorithms

This research work performed multi-objective optimization of MRR and TWR for PSS with three different values (i.e., 12.69%, 18.85%, and 31.11%) using TLBO and PSO. The outcome is updated in both stages of the TLBO approach, i.e., the teacher and the learner phases. In addition, only two algorithm-specific parameters exist: size of population and generation number. However, in PSO, extra algorithm-specific parameters such as inertia coefficient, personal acceleration coefficient, and social acceleration coefficient must also be specified. Based on many iterations, the population size and generation number for both the TLBO and PSO algorithms have been determined to be 9 and 50, respectively. The constant parameters in PSO are, the inertia coefficient is set to 0.7, the personal acceleration coefficient is set to 1.5, and the social acceleration coefficient is set to 1.5. Table 9 shows the results of utilizing PSO and TLBO to optimize the MRR and TWR for EDM of PSS. MRR (Exp.) and TWR (Exp.) show the experimental values of MRR. MRR (Pred.) and TWR (Pred.) show the predicted values of TWR obtained from the regression equations, i.e., Equations (4)–(9). It is essential in the TLBO algorithm that the final output result is revised in the teacher and learner phases. Personal best and global best values define the PSO algorithm’s performance rate. In the case of PSS having 12.69% porosity, the PSO and TLBO algorithm provides the same MRR(Exp.) value of 0.3022 mm${}^{3}$/s and the same TWR value of 0.3237 mg/s, as shown in Table 9. In the second case of PSS having 18.85% porosity, the PSO algorithm provides the MRR(Exp.) value of 0.2517 mm${}^{3}$/s and the TWR(Exp.) value of 0.3066 mg/s, and the TLBO algorithm provides the MRR(Exp.) value of 0.2384 mm${}^{3}$/s and the TWR(Exp.) value of 0.3222 mg/s, as shown in Table 9. Hence, the PSO algorithm improves by about 5.58% in MRR(Exp.) and 4.84% in TWR(Exp.) over the TLBO. Thus, the PSO algorithm provides better MRR(Exp.) and TWR(Exp.) values than TLBO. In the third case of PSS having 31.11% porosity, the PSO algorithm provides an MRR(Exp.) value of 0.2005 mm${}^{3}$/s and a TWR(Exp.) value of 0.2855 mg/s, and the TLBO algorithm provides an MRR(Exp.) value of 0.1924 mm${}^{3}$/s and a TWR value of 0.3026 mg/s, as shown in Table 9. Hence, the PSO algorithm improves by about 4.21% in MRR(Exp.) and 5.65% in TWR (Exp.) over the TLBO. Thus, the PSO algorithm provides better MRR and TWR values than TLBO.

The convergence of graphs of TLBO and PSO for three different cases is shown in Figure 8. In the first case, PSS with a porosity value of 12.60%, the convergence curve shows that the best fitness value for PSO and TLBO is the same, i.e., 0.5403, and this value is obtained after 20 iterations for TLBO and 3 iterations for PSO, as shown in Figure 8a. In the second case, PSS with a porosity value of 18.85%, the convergence curve shows that the best fitness value for PSO is 0.5682 and for TLBO is 0.5069, and this value is obtained after 21 iterations for TLBO and 2 iterations for PSO, as shown in Figure 8b. In the third case, PSS with a porosity value of 31.11%, the convergence curve shows that the best fitness value for PSO is 0.5787 and for TLBO is 0.5239, and this value is obtained after 19 iterations for TLBO and 2 iterations for PSO, as shown in Figure 8c. Hence, from the perspective of convergence rate, the PSO algorithm is found to be consistent and converged more quickly, taking the minimum computational time and effort compared to the TLBO algorithm.

### 4.5. Confirmation Experimentation

In any DOE experimental method, the confirmation experiment is the last stage. The confirmation experiment’s objective is to verify the results established during the analysis process. The machining parameters’ optimal levels were used to predict and validate the performance measure’s improvement. To calculate the MRR and TWR values, a new experiment was developed using a combination of factors and their levels, as presented in Table 9. The highest percentage of relative errors associated with MRR and TWR was found to be 4.37% and 4.73% in the empirically confirmed optimum conditions. These figures are suitable from the standpoint of the engineering discipline and ensure the success of the statistical design technique adopted.

### 4.6. SEM Results

The morphology variation of sintered samples is characterized by using SEM analysis. The morphology of PSS consisting of 10 wt.%, 20 wt.%, and 30 wt.% NaCl is shown in Figure 9a–c. Pores are visible in such samples. As we have already mentioned, the machining of PSS by conventional techniques is impossible, as this phenomenon results in blockage of pores, as shown in Figure 9d. PSS can only be machined by the EDM process. Improper selection of EDM process parameters leads to melting of PSS, as shown in Figure 9e, which results in blockage of pores. Therefore, there is a need to optimize the EDM process parameters so that efficient machining can be obtained without disturbing the inter-connectivity of the pores and no pore blockage, as shown in Figure 9f.

## 5. Conclusions

The present research work was performed to investigate the effect of EDM process parameters on the machining of PSSs with different porosity. Meta-heuristic techniques such as TLBO and PSO algorithms were utilized to obtain accuracy in multiple responses, i.e., MRR and TWR values. The EDM of porous metal requires optimal process parameters, which significantly benefit manufacturing industries in terms of retained breathing capacity (or interconnected pores), improved product quality, reduced machining cost, enhanced productivity, and reduced experimentation cost, time, and error. Hence, the following conclusions can be drawn based on experimental work and the analysis of optimization results.

With increasing porosity values of PSS, the average MRR values decreased by 17.16% and further decreased by 21.67%.With increasing porosity values of PSS, the average TWR values decreased by 6.26% and further decreased by 9.19%.The optimum machining parameters for PSS with a 12.60% porosity value were obtained as $\left(T_{on}\right)$ 15 μs, $I_{p}$ 10 A, *V* 25 V, and $f_{p}$ 1 kg/cm${}^{3}$ for both TLBO and PSO algorithms.The optimum machining parameters for PSS with an 18.85% porosity value were obtained as $\left(T_{on}\right)$ 15 μs, $I_{p}$ 10 A, *V* 25 V, and $f_{p}$ 1 kg/cm${}^{3}$ for the PSO algorithm, and $\left(T_{on}\right)$ 15 μs, $I_{p}$ 10 A, *V* 20 V, and $f_{p}$ 1 kg/cm${}^{3}$ for the TLBO algorithms.The optimum machining parameters for PSS with a 31.11% porosity value were obtained as $\left(T_{on}\right)$ 15 μs, $I_{p}$ 10 A, *V* 25 V, and $f_{p}$ 1 kg/cm${}^{3}$ for the PSO algorithm, and $\left(T_{on}\right)$ 15 μs, $I_{p}$ 10 A, *V* 20 V, and $f_{p}$ 1 kg/cm${}^{3}$ for the TLBO algorithms.In the case of PSS with an 18.85% porosity value, the PSO algorithm improves by about 5.25% in MRR and by 5.63% in TWR over the TLBO.In the case of PSS with a 31.11% porosity value, the PSO algorithm improves by about 3.73% in MRR and by 6.46% in TWR over the TLBO.The PSO algorithm is found to be consistent and to converge quicker, taking minimal computational time and effort compared to the TLBO algorithm.

## Figures and Tables

**Figure 1 materials-15-06571-f001:**
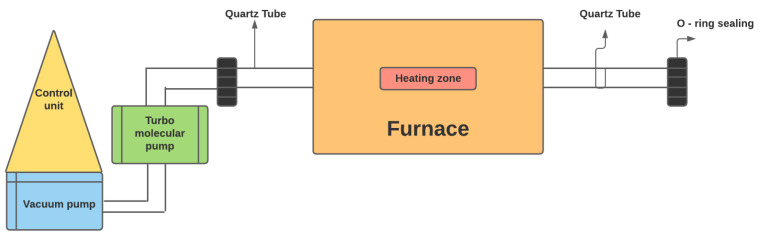
Turbo-molecular pump assisted vacuum furnace setup.

**Figure 2 materials-15-06571-f002:**
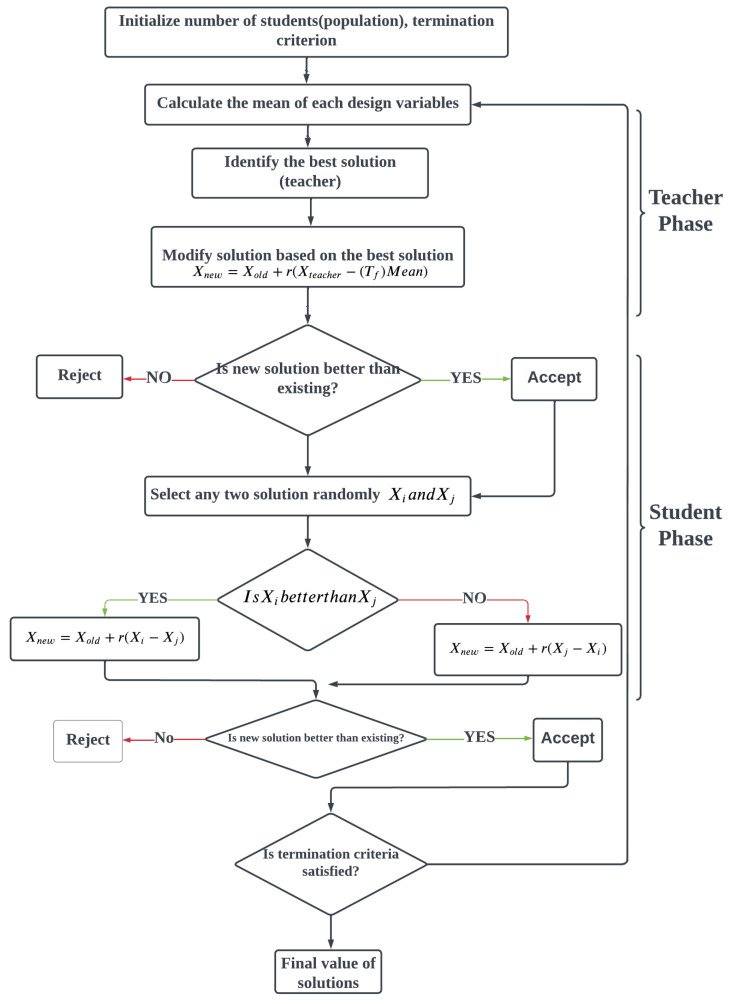
Flow chart for Teaching–Learning-Based Optimization (TLBO).

**Figure 3 materials-15-06571-f003:**
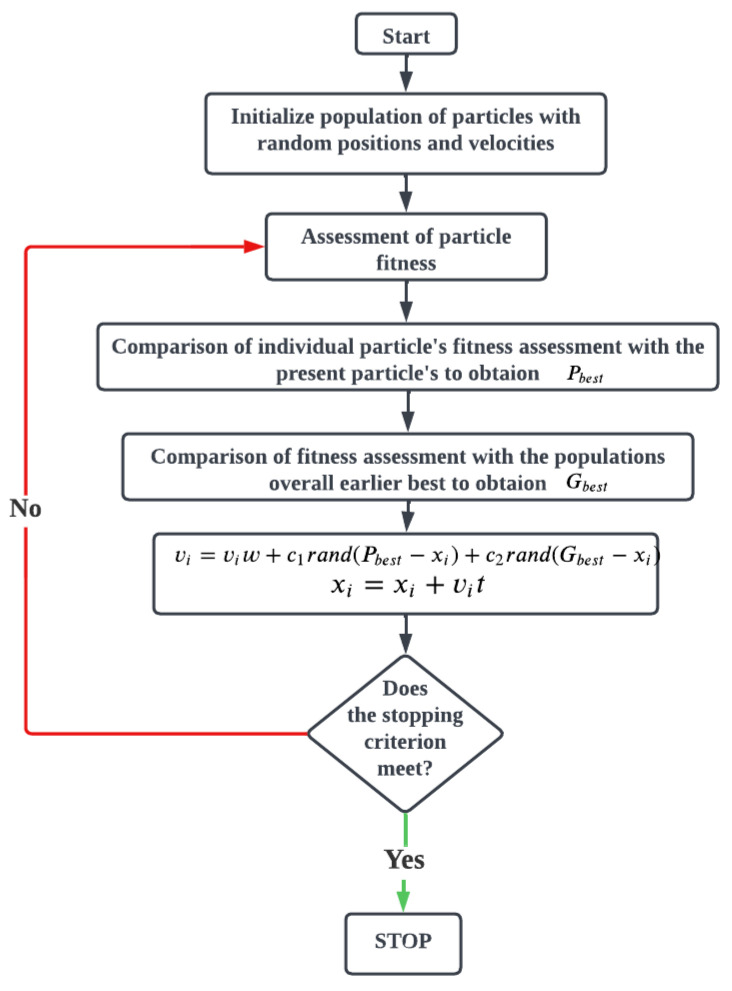
Flow chart for Particle Swarm Optimization (PSO).

**Figure 4 materials-15-06571-f004:**
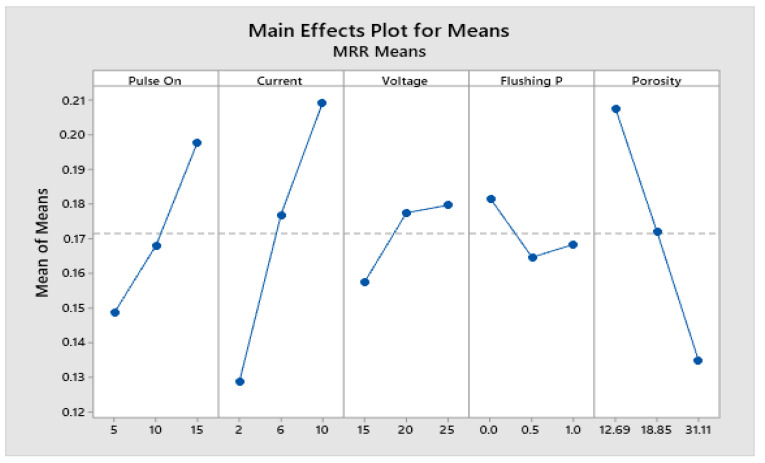
Main effect plot of factors for MRR.

**Figure 5 materials-15-06571-f005:**
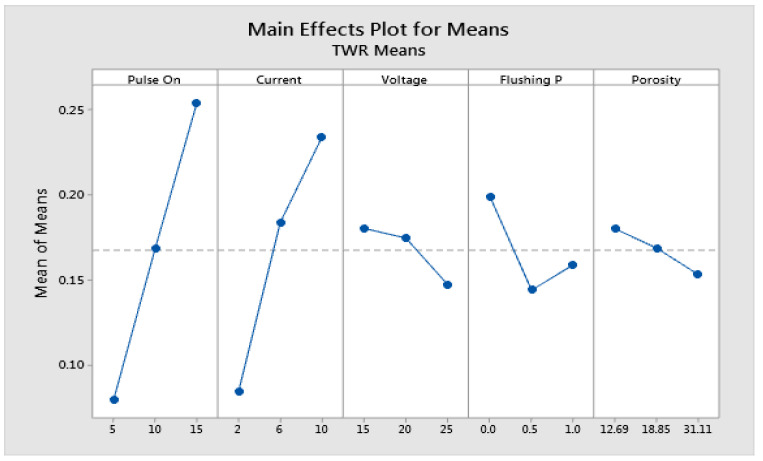
Main effect plot of factors for TWR.

**Figure 6 materials-15-06571-f006:**
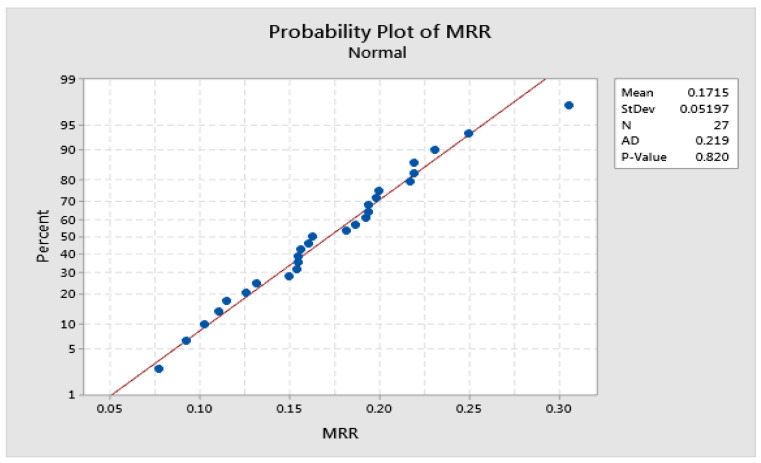
Normal probability plot of residuals for MRR.

**Figure 7 materials-15-06571-f007:**
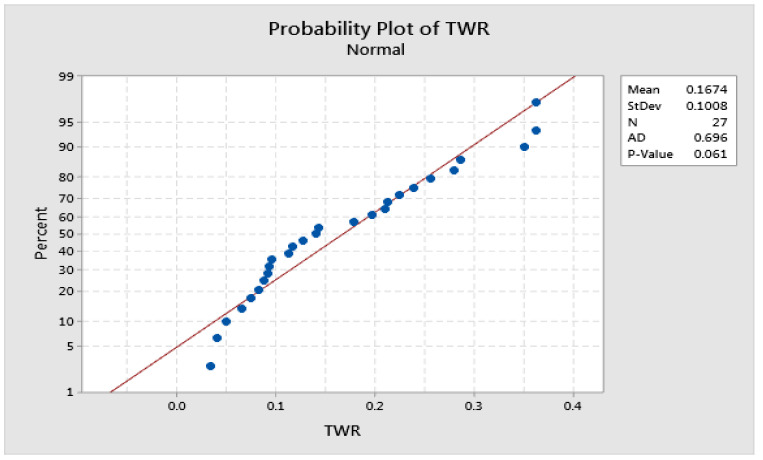
Normal probability plot of residuals for TWR.

**Figure 8 materials-15-06571-f008:**
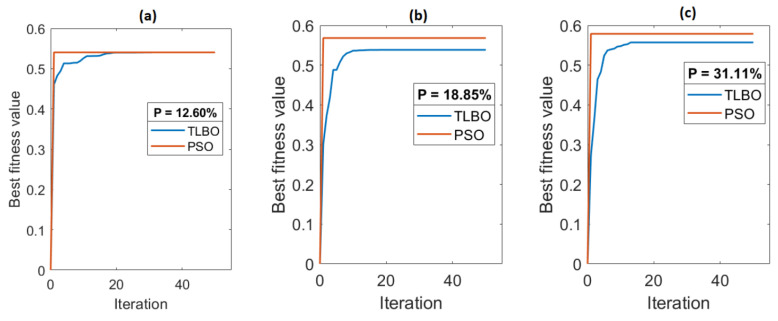
Convergence curves of PSO and TLBO for optimization of MRR and TWR. (**a**) convergence curve or 12.60% PSS, (**b**) convergence curve for 18.85% PSS, (**c**) convergence curve for 31.11% PSS.

**Figure 9 materials-15-06571-f009:**
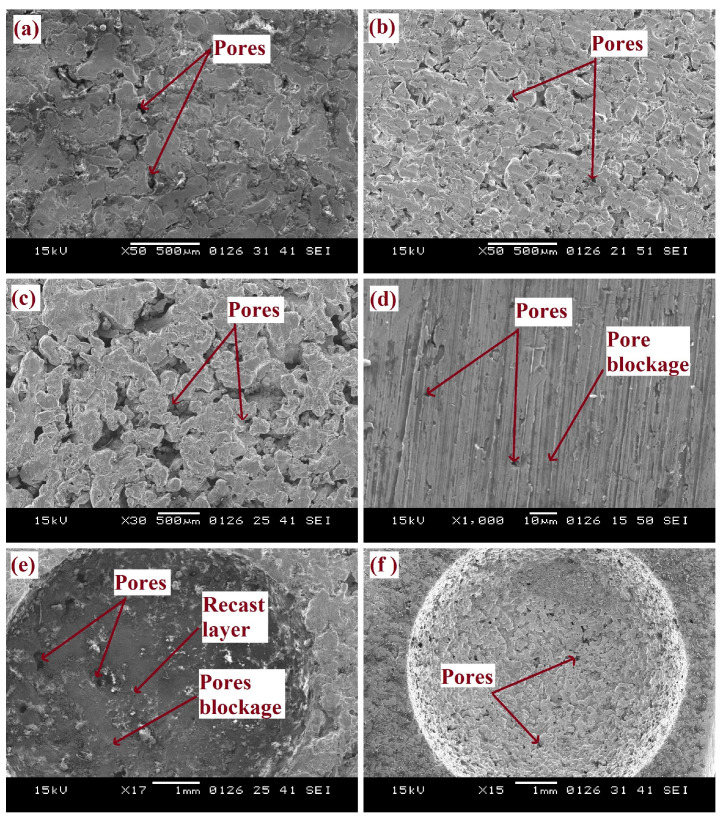
Porous stainless steel morphology: (**a**) 10 wt.% NaCl, (**b**) 20 wt.% NaCl, (**c**) 30 wt.% NaCl, (**d**) Conventional machining, (**e**) Pores blockage by EDM, and (**f**) EDM process at optimum parameters.

**Table 1 materials-15-06571-t001:** Chemical composition of SS316L metal powder.

C	Si	Mn	Cr	Ni	Mo	P	S	Fe
0–0.03	0–1	0–2	16–18	10–12	2–3	0–0.04	0–0.03	Balance

**Table 2 materials-15-06571-t002:** Input Process Parameters and Their Levels.

Process Parameter	Level 1	Level 2	Level 3
Peak Current (A)	2	6	10
Pulse On Time (μs)	5	10	15
Flushing Pressure (kg/cm${}^{2}$)	0	0.5	1
Voltage (V)	15	20	25
Porosity (%)	12.69	18.85	31.11

**Table 3 materials-15-06571-t003:** Values of fixed input parameter.

Parameter	Depth	Duty Cycle	Bi Pulse Current	Spark Time	Lift
Value	2 (mm)	8	3 (A)	6 (μs)	0.8 (mm)

**Table 4 materials-15-06571-t004:** Experimetal layout using an $L_{27}$ Orthogonal array.

Exp No	Pulse on Time	Current	Voltage	Flushing Pressure	Porosity	MRR	TWR
1	5	2	15	0	12.69	0.1255	0.0499
2	5	2	15	0	18.85	0.1023	0.0409
3	5	2	15	0	31.11	0.0769	0.0340
4	5	6	20	0.5	12.69	0.1936	0.0916
5	5	6	20	0.5	18.85	0.1548	0.0831
6	5	6	20	0.5	31.11	0.1105	0.0652
7	5	10	25	1	12.69	0.2307	0.1400
8	5	10	25	1	18.85	0.1940	0.1165
9	5	10	25	1	31.11	0.1495	0.0959
10	10	2	20	1	12.69	0.1606	0.0927
11	10	2	20	1	18.85	0.1311	0.0871
12	10	2	20	1	31.11	0.0918	0.0744
13	10	6	25	0	12.69	0.2194	0.2133
14	10	6	25	0	18.85	0.1924	0.1968
15	10	6	25	0	31.11	0.1627	0.1792
16	10	10	15	0.5	12.69	0.2189	0.2397
17	10	10	15	0.5	18.85	0.1814	0.2247
18	10	10	15	0.5	31.11	0.1537	0.2098
19	15	2	25	0.5	12.69	0.1983	0.1432
20	15	2	25	0.5	18.85	0.1561	0.1273
21	15	2	25	0.5	31.11	0.1144	0.1127
22	15	6	15	1	12.69	0.2170	0.2873
23	15	6	15	1	18.85	0.1865	0.2804
24	15	6	15	1	31.11	0.1543	0.2569
25	15	10	20	0	12.69	0.3057	0.3636
26	15	10	20	0	18.85	0.2501	0.3629
27	15	10	20	0	31.11	0.1993	0.3517

**Table 5 materials-15-06571-t005:** Analysis of variance Analysis for MRR.

Source	DF	Adj SS	Adj MS	F-Value	*p*-Value	%Contri.
Regression	5	0.0665	0.0133	76.0000	0.0000	94.7629
Pulse On	1	0.0110	0.0110	62.5500	0.0000	15.5973
Current	1	0.0293	0.0293	167.4000	0.0000	41.7442
Voltage	1	0.0022	0.0022	12.8100	0.0020	3.1961
Flushing P	1	0.0008	0.0008	4.5100	0.0460	1.1252
Porosity	1	0.0232	0.0232	132.7400	0.0000	33.1002
Error	21	0.0037	0.0002			
Total	26	0.0702				

**Table 6 materials-15-06571-t006:** Model summary of regression analysis for MRR.

S	R-sq	R-sq (adj)	R-sq (pred)
0.0132	94.76%	93.52%	91.28%

**Table 7 materials-15-06571-t007:** Analysis of variance analysis for TWR.

Source	DF	Adj SS	Adj MS	F-Value	*p*-Value	%Contri.
Regression	5	0.2523	0.0505	87.2500	0.0000	95.4072
Pulse On	1	0.1367	0.1367	236.4100	0.0000	51.7044
Current	1	0.1001	0.1001	173.1300	0.0000	37.8643
Voltage	1	0.0050	0.0050	8.5800	0.0080	1.8761
Flushing P	1	0.0072	0.0072	12.5300	0.0020	2.7406
Porosity	1	0.0032	0.0032	5.5900	0.0280	1.2219
Error	21	0.0121	0.0006			
Total	26	0.2644				

**Table 8 materials-15-06571-t008:** Model summary of regression analysis for TWR.

S	R-sq	R-sq (adj)	R-sq (pred)
0.0240	95.41%	94.31%	92.51%

**Table 9 materials-15-06571-t009:** Optimization results using TLBO and PSO for MRR and TWR.

No.	Poro.	Tech.	$T_{\mathrm{on}}$	I	V	$F_{p}$	MRR (Pred.)	MRR (Exp.)	% Err.	TWR (Pred.)	TWR (Exp.)	% Err.
1.	12.69	PSO	15	10	25	1	0.2890	0.3022	4.37	0.3106	0.3237	4.05
		TLBO	15	10	25	1	0.2890	0.3022	4.37	0.3106	0.3237	4.05
2.	18.85	PSO	15	10	25	1	0.2416	0.2517	4.01	0.2949	0.3066	3.82
		TLBO	15	10	20	1	0.2295	0.2384	3.73	0.3125	0.3222	3.01
3.	31.11	PSO	15	10	25	1	0.1930	0.2005	3.74	0.2720	0.2855	4.73
		TLBO	15	10	20	1	0.1861	0.1924	3.27	0.2908	0.3026	3.90

## Data Availability

Data is contained within the article.

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
