# Peer review of "Multi-Objective Optimization of the Process Parameters in Electric Discharge Machining of 316L Porous Stainless Steel Using Metaheuristic Techniques"

_materials, 2022, doi:10.3390/ma15196571_

Round 1

Reviewer 1 Report

The reviewer comments of the paper submitted to Materials
«Multi-Variable Optimization of the Process Parameters in Die-Sinking Electric Discharge Machining of 316L Porous Stainless Steel»
- Reviewer
The authors presented an article «Multi-Variable Optimization of the Process Parameters in Die-Sinking Electric Discharge Machining of 316L Porous Stainless Steel». The article is interesting and deserves the attention of readers. However, there are several points in the article that require further explanation. The authors need to perform all changes according to comments one-by-one.
Comment 1:
Title needs to be concretized. What exactly is explored in the article? By what methods?

Comment 2:
The abstract needs to be improved.
Demonstrate in the abstract novelty, practical significance. Add quantitative and qualitative work results to the abstract. The obtained findings need to be put in the numerical form.
Comment 3:
The introduction needs to be improved.
Group quotation is unacceptable in one phrase, for example [1-3]. Break this sentence into parts or individual sentences. For example, ... [...], ... [...], etc. Or one reference - one sentence.
Now the list of references needs to be supplemented with at least 6-7 more the latest review and research articles.

Comment 4:
After analyzing the literature, show before formulating the goal of the "blank" spots. Which has not been previously done by other researchers. You must show the importance of the research being undertaken. Show what will be the new research approach in this article. You need to show a hypothesis.

Comment 5:
Add scientific novelty and practical relevance.
Add a clear purpose to the article.
Comment 6:
3. Design and construction
Are all figures original? If not needed appropriate citations and permissions. Refine this for figures throughout the article.
Are all formulas original? If not needed appropriate citations.
Comment 7:
4. Testing
Add the material chemistry of the stock in a separate table. How was the material hardness measured?
How the authors selected the test conditions? Please describe it in depth.
Comment 8:
Describe the measurement procedure in more detail. At what point in time? How is the measuring setup set up? How many repetitions of measurements? What statistical methods are used to process experimental results? Describe the experimental stand in more detail. What method of experiment planning is used and why?
Comment 9: Results and discussions need to be improved. I see that there are explanations without adding any references in this section. Also, the explanations have to be improved considering the all effects of parameters. Please consider the following papers for the better analysis of ANOVA:

Parametric optimization for cutting forces and material removal rate in the turning of AISI 5140

Anova and fuzzy rule based evaluation and estimation of flank wear, temperature and acoustic emission in turning

Comment 10:
Conclusions should be improved.
It is necessary to more clearly show the novelty of the article and the advantages of the proposed method. What is the difference from previous work in this area? Show practical relevance.
The article is interesting, but needs to be improved. Authors should carefully study the comments and make improvements to the article step by step. After minor changes can an article be considered for publication in the "Materials".

Reviewer 2 Report

This paper optimized the EDM process of porous stainless steel by Taguchi orthogonal method, TLBO and PSO. Maybe, it could be published after addressing the following questions:

(1) The introduction is tedious and lacking logical hierarchy.

(2) The main text needs to be substantially cut down. For example, the reason behind the selection of copper as an electrode material and the introduction of DOE, TLBO and PSO should be cut off.

(3) In 2.2. Sintering of SS316L Metal Powder, the vacuum atmosphere is up to 10-5 mbar, namely 10-8 bar?

(4) What kind of copper is the electrode?

(5) The conclusion should be more succinct.
